# Plant Growth and Chemical Properties of Commercial Biochar- versus Peat-Based Growing Media

Bruno Glaser *[ID] and Angela Amma Asieduaa Asomah

Soil Biogeochemistry, Institute of Agricultural and Nutritional Sciences, Martin Luther University Halle-Wittenberg, 06120 Halle (Saale), Germany; angela.asomah@posteo.de
* Correspondence: bruno.glaser@landw.uni-halle.de

**Abstract:** Peatlands have been irreversibly destroyed by draining and mining for horticulture, in the course of which tremendous amounts of greenhouse gasses were released into the atmosphere. To avoid this in the future, sustainable alternatives are urgently needed to substitute peat as growing media. An appropriate alternative could be biochar, because it has beneficial effects on nutrient availability and retention, water holding capacity, and organic matter stability. In this study, we compared three different commercially available biochar-containing growing media (Palaterra, Sonnenerde, Terra Magica) with three commercially available peat-based growing media (CompoSana, Dehner die leichte, Dehner mit Vorratsdünger), in a randomized greenhouse pot experiment. Pure sand was used as a control and, to test a potential amount effect, we mixed the used growing media with increasing amounts of pure sand (0, 25, 50, 75, and 100 volume % of individual growing media). The consecutive yields of several agronomically relevant cereals (barley, wheat, and maize) were measured in the mixtures mentioned previously. Additionally, the contents of biochar, amino sugar, and polycyclic aromatic hydrocarbons were measured in each pure growing media before and after the growth experiments. Only Sonnenerde exhibited an increased plant yield of 30–40% compared with peat-based growing media. The growing media exhibited no significant differences of chemical soil properties during the experiment. Only slight tendencies are recognizable towards higher fungal community in biochar- and peat-based growing media. A clear fungi contribution was observed in Palaterra, most probably due to the fact that fungi was a production ingredient. Surprisingly, peat-based growing media also contained about 30 g kg$^{-1}$ black carbon, a polycondensed aromatic carbon typical for biochar. Overall, our results indicated that biochar-containing growing media, especially Sonnenerde, is a potential alternative for peat-based growing media in horticulture and can enhance degraded soils.

**Keywords:** commercial biochar-containing growing media; Sonnenerde; Terra Magica; Palaterra; peat replacement in growing media

## 1. Introduction

About 4 million km$^2$ worldwide are covered by peatland, equivalent to 3% of the total land surface containing about 44% of total terrestrial carbon [1]. In Central Europe, large areas of peatlands have already been extracted. Therefore, the region depends on the supply of peat from the Baltic States in order to maintain or expand its horticultural production. Total peat extraction is estimated at 25 million m$^3$ per year for Finland, 15–20 million m$^3$ per year for Estonia, 9–11 million m$^3$ per year for Ireland, and 7 million m$^3$ per year for Germany [1]. The amount of peat imported from Northern and Eastern Europe to Germany will increase from 1–2 million m$^3$ per year to 7–8 million m$^3$ per year in the next 20 years [1]. On the other hand, peat is a non-renewable resource which degrades via oxidation when in contact with air, and its stored carbon is released into the atmosphere as climate-relevant carbon dioxide. The results of drainage are devastating to peat bogs for a number of reasons, including the loss of biodiversity, the increase in carbon emissions, and the effects

of climate change. Peat has a large water-holding capacity, a good air-filled porosity, a low pH, low bulk density, low nutrient content, and a useful cation exchange capacity, and these unique properties make it suitable as a raw material in the growing media sector [2]. The low pH and nutrient content can be adjusted to meet specific plant requirements.

It is essential to ensure a sustainable use of natural resources and to prevent the mining of peatland because this is one of the ecosystems necessary for the regulation of local water quality, water regime, and flood protection [3]. Environmentalists, governments, and horticultural businesses are searching for alternative growing media to act as a substitute for peat, so as to avoid the mining of bogs, as they recognize the environmental consequences. One potential alternative to replace peat in horticulture systems and agriculture is biochar or terra preta substrates, because biochar is a porous material with a high surface area, which is responsible for its positive chemical, physical, and microbial properties [1,2,4].

Biochar has the potential to increase crop yields, plant nutrient availability, soil water availability, microbial biomass, and soil microbial diversity [5].

Therefore, the aims of this study were (i) to quantify the short-term effect of three different commercially available biochar- and peat-based growing media on plant growth and chemical properties, and (ii) to assess a potential amount effect when using those growing media for dilution with existing soil. We hypothesize that commercially available biochar-based growing media are suitable alternatives for peat-based growing media.

## 2. Material and Methods

A pot experiment was conducted at a greenhouse of the Department of Soil Bio-geochemistry at the University of Halle, Germany. Six different commercially available growing media were tested, three commercially available peat-based growing media and three commercially available biochar-based growing media. An overview of each growing media is given in Table 1. As an additional control, we used pure sand. To better detect the growth potential, each growing media was diluted with pure sand in increasing amounts. Each treatment was independently replicated 5 times, and pots were organized randomly within the greenhouse. In total, 45 polyvinyl chloride pots (15 cm in diameter and 40 cm in height) were supplied with drainage holes at the bottom. For each consecutive experiment, there was a growing period of thirty days for three following agronomically relevant plants: barley (*Hordeum vulgare*), wheat (*Triticum aestivum*), and maize (*Zea mays*). The watering (1 min every 6 h) and illumination (40 klux from 8 am to 8 pm) were constant throughout the growing periods.

**Table 1.** Ingredients and production processes of commercially available peat- and biochar-based growing media used in this study according to information of suppliers.

| Commercial Growing Media | Ingredients | Production Process |
|---|---|---|
| **Peat-Based** | | |
| Compo Sana | 95% Bog peat (H2 - H7) [1], NPK fertilizer, wetting agent, perlite, lime, phosphate with silica (Agrosil) | Only mixing |
| Dehner die Leichte | 98% Bog peat (H2 - H7) [1], NPK fertilizer | Only mixing |
| Dehner mit Vorratsdünger | 92% Bog peat (H3 - H7) [1], NPK fertilizer, perlite, lime seabird guano (0.07%) | Only mixing |
| **Biochar-Based** | | |
| Palaterra | Green waste compost, wood fibre, bark humus, biochar, bentonite, basalt, bugle fertilizer, microorganism, fungi | Aerobic rotting, fermentation 1, fermentation 2 |
| Sonnenerde | Green waste compost, fruit waste, rock flour, biochar, mycorrhiza, N-binding bacteria | Aerobic composting |
| Terra Magica | Biomass, biochar, microorganisms rock flour, | Aerobic composting |

[1] H2 - H7 is the degree of decomposition. H2: white peat readily decomposed, H7: black peat strongly decomposed.

After 30 days, all plants were cut, collected in brown paper bags and dried in an oven at 105 °C for 3 days. The growing media of each treatment were collected at the beginning and end of all three growth periods. Total organic carbon (TOC) and total nitrogen (TN) of

the soil samples were determined after combustion at 1050 °C by thermal conductivity on a Vario EL elemental analyzer. Plant-available (exchangeable) cations Ca, Mg, K, and Na were analyzed using ICP.

The sum of the set of 16 polycyclic aromatic hydrocarbons, proposed by the Environmental Protection Agency for the prevention of health threats in the commercially available growing media, were extracted via Soxhlet extraction using toluene, followed by gas chromatographic separation and mass spectrometric identification, and quantification by a commercial laboratory (Synlab, Leizpig, Germany).

Black carbon was analyzed in the commercially available growing media to detect the polycondensed aromatic moieties of biochar using benzene polycarboxylic acids (BPCA) as molecular markers, following the method of Glaser et al. [6]. Elimination of polyvalent cations was done with 4 M trifluoroacetic acid [7]. Production of BPCA was carried out by oxidation with 65% $HNO_3$ for 8 h at 170 °C at elevated pressure. After subsequent sample cleanup and derivatization of BPCA, identification and quantification of individual BPCA was carried out using gas chromatography with flame ionization detection [6].

Amino sugars in the commercially available growing media were determined according to Zhang and Amelung [8]. For this purpose, growing media samples containing about 0.5 mg N were hydrolyzed with 6 M HCL for 8 h at 105 °C to release amino sugars from the mineral-organic matrix. After sample-cleanup, especially alkaline precipitation of iron, derivatization, identification, and quantification of individual amino sugars was carried out using gas chromatography and flame ionization detection.

Data were statistically analyzed using SPSS® 23 and Excel 2010. Data were tested for normality using the Kolmogorov–Smirnov test, and for homogeneity using the Levene test. Significant difference between the biomass measurements in the different mixture samples were tested with Spearman's correlation.

## 3. Results

### 3.1. Biomass Yields

#### 3.1.1. Barley

Barley yield ranged between 0.9 and 4.7 g per pot, with no significant difference between biochar- and peat-containing growing media mixed with sand (Figure 1). Apart from Palaterra, barley yields were homogeneous among biochar- and peat-containing growing media (Figure 1). Surprisingly, biochar-based growing media (Terra Magica, Sonnenerde) and CompoSana mixed with 50% sand demonstrated a better yield than in the pure growing media.

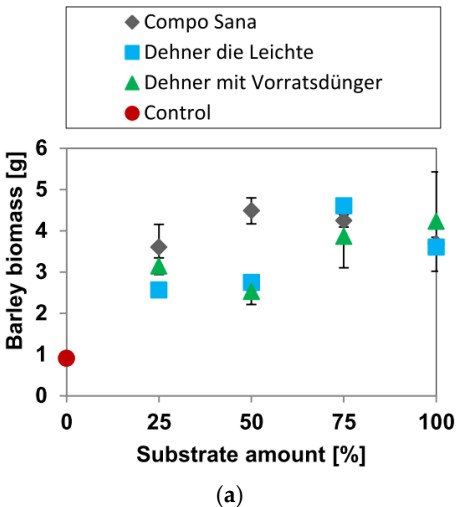
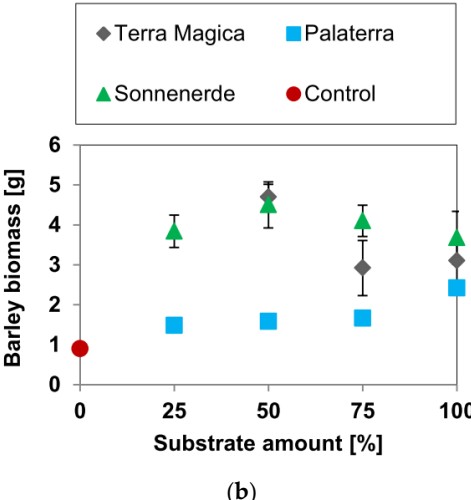

(**a**)          (**b**)

**Figure 1.** Barley yield per pot after 30 days in peat-based (**a**) and biochar-based (**b**) growing media mixed with sand in different amounts. In this Figure, 0% = control (pure sand) and 100% = pure growing media. Error bars represent standard error of the mean (n = 5).

### 3.1.2. Wheat

Wheat biomass ranged between 0.8 and 4.0 g per pot with partially significant quantity effects between biochar- and peat-containing growing media (Figure 2). Only Sonnenerde produced a higher wheat biomass compared to pure sand, with increasing yields the more Sonnenerde substrate was used ($R^2$ = 0.92; Figure 2). All other growing media exhibited a neutral or even negative amount effect, and wheat yield was not higher than in pure sand (Figure 2). The wheat yields from Dehner die Leichte und Dehner mit Vorratsdünger were comparable with Terra Magica and Palaterra. Wheat yield in the pure growing media decreased in the following order: Sonnenerde > Compo Sana > Dehner die Leichte ≈ Dehner mit Vorratsdünger > Palaterra ≈ Terra Magica.

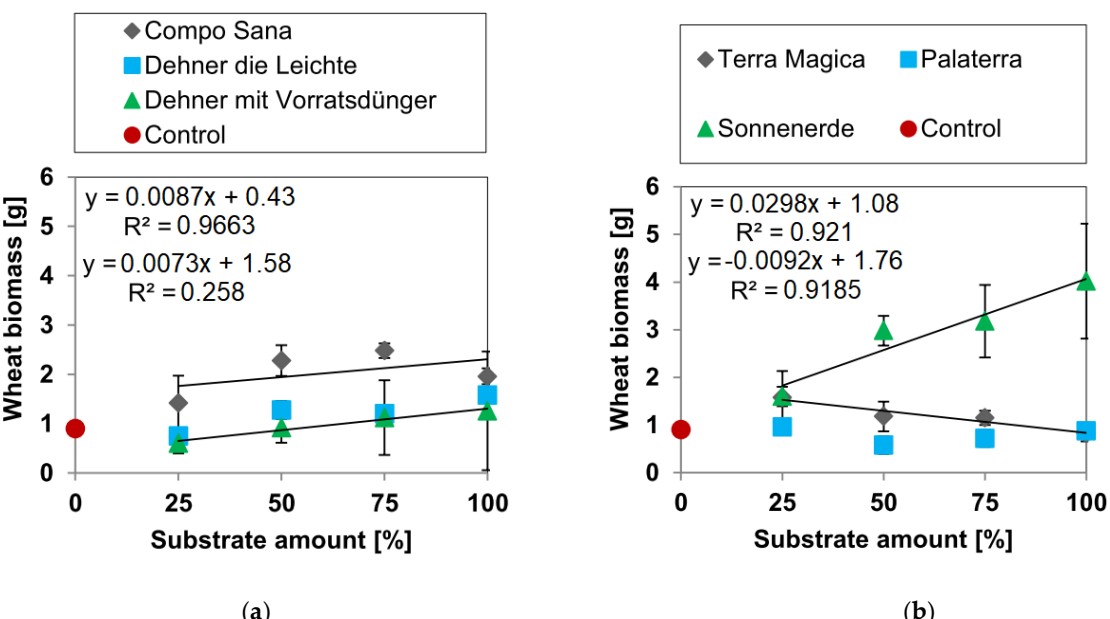

(**a**)  (**b**)

**Figure 2.** Wheat yield per pot after 30 days in peat-based (**a**) and biochar-based (**b**) growing media mixed with sand in different amounts. In this Figure, 0% = control (pure sand) and 100% = pure growing media. Errors bars represent standard error of the mean (n = 5). $R^2$ was calculated with the Spearman correlation coefficient.

### 3.1.3. Maize

Maize yield ranged between 0.6 and 11.1 g per pot, with partially significant quantity effects among biochar- and peat-based growing media (Figure 3). Peat-containing growing media such as Dehner die Leichte and Dehner mit Vorratsdünger did not differ from each other, and only Compo Sana exhibited a higher yield than the other peat-based growing media or pure sand. Apart from Sonnenerde ($R^2$ = 0.76), biochar-containing growing media exhibited a negative substrate amount effect and were somewhat better than the control. Maize yield in the pure growing media decreased in the following order: Sonnenerde > Compo Sana > Dehner mit Vorratsdünger > Dehner die Leichte > Palaterra. Maize yield in the pure Terra Magica growing media could not be measured.

### 3.1.4. Total Biomass Yield

Total biomass yield ranged between 3.9 and 18.3 g per pot, with a significant substrate amount effect among all peat-based growing media and only for one biochar-containing growing media (Sonnenerde, Figure 4). Total biomass yield of peat-based growing media significantly increased with increasing amount of growing media (Figure 4a). For the biochar-based growing media, only Sonnenerde exhibited a significant total biomass yield increase with increasing growth media amount (Figure 4b). The other two biochar-based

growing media showed total biomass yields comparable with peat-based growing media being independent from the amount of growing media used (Figure 4).

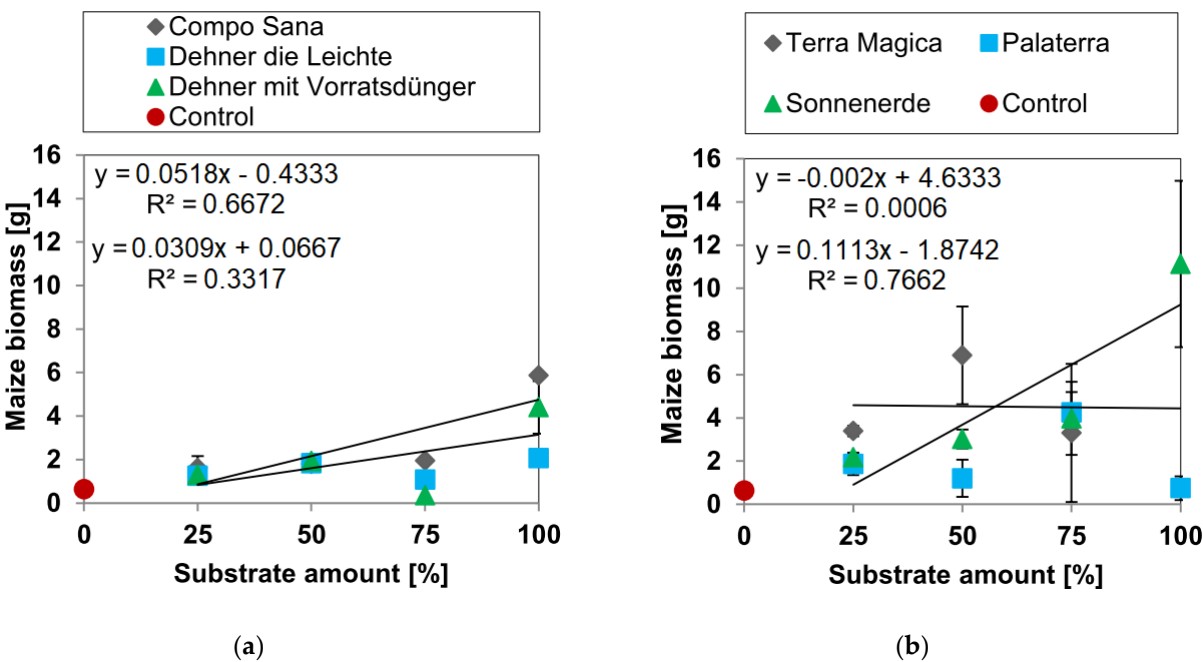

(**a**) (**b**)

**Figure 3.** Maize yield per pot after 30 days in peat-based (**a**) and biochar-based (**b**) growing media mixed with sand in different amounts. In this Figure, 0% = control (pure sand) and 100% = pure growing media. Error bars represent standard error of the mean (n = 5). $R^2$ was calculated with the Spearman correlation coefficient.

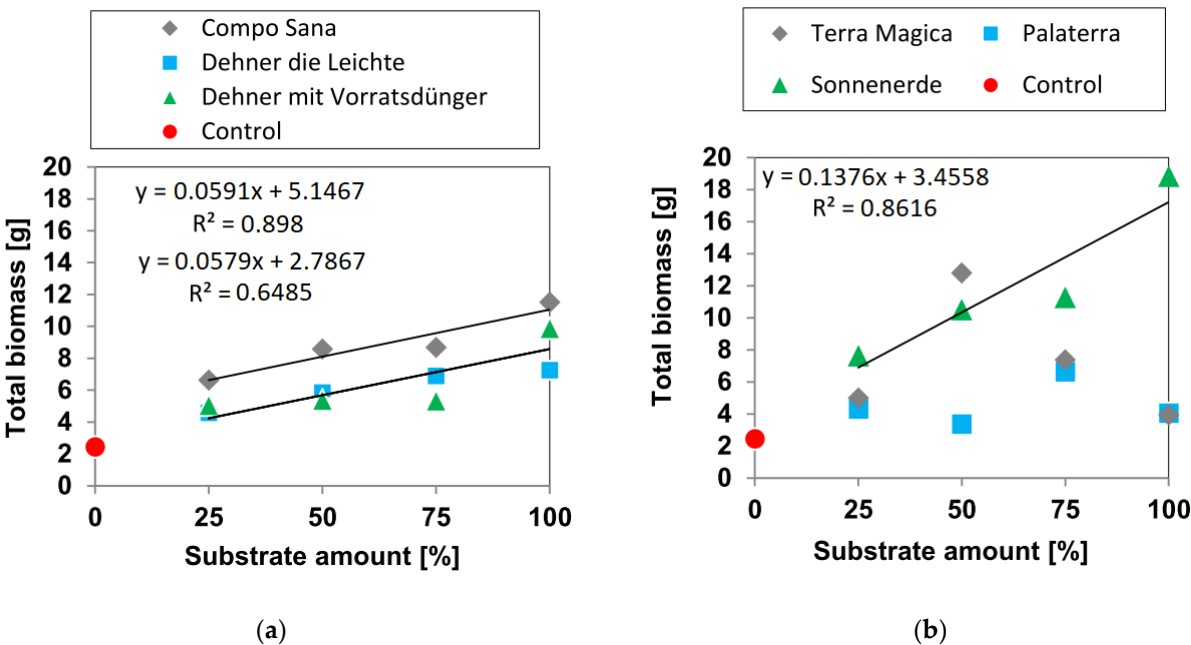

(**a**) (**b**)

**Figure 4.** Total biomass yield (Barley plus wheat plus maize) per pot after 90 days in peat-based (**a**) and biochar-based (**b**) growing media mixed with sand in different amounts. In this Figure, 0% = control (pure sand) sand 100% = pure growing media. $R^2$ was calculated with the Spearman coefficient.

### 3.2. Plant-Available Nutrient Stocks and Nutrient Retention

Plant-available nutrient content ranged between 15 and 79 g kg$^{-1}$, being significantly higher in biochar-containing compared to peat-based growing media (Figure 5). Nutrient content decreased in the following order Sonnenerde > Terra Magica > Palaterra > Compo Sana ≈ Dehner die Leichte > Dehner mit Vorratsdünger > Control (Figure 5). In all treatments, the plant-available nutrient content decreased by 5–46% during the three growing periods. Nutrients were dominated by calcium in all growth media ranging between 29–46 g kg$^{-1}$, covering more than 50% of all nutrient content, followed by potassium, magnesium, iron, and phosphate (Figure 5). Generally, two (Sonnenerde and Terra Magica) out of three biochar-based growing media exhibited higher nutrient content compared to the peat-based growing media. Only Palaterra released the same nutrient content as peat-containing growing media. Nutrient loss of all investigated growing media were relatively low after three vegetation periods (Figure 5).

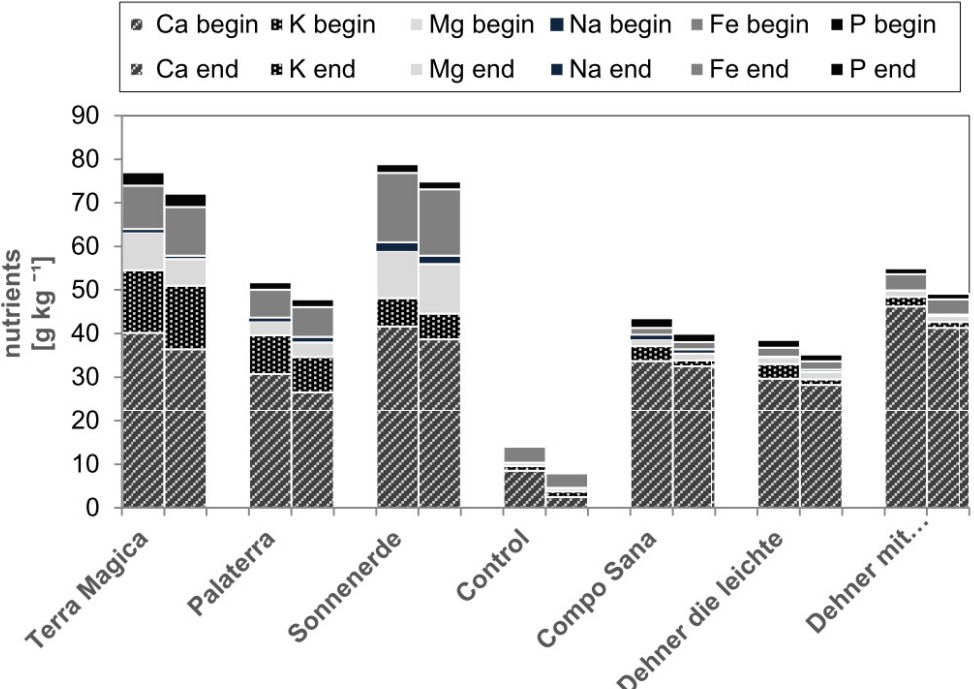

**Figure 5.** Plant-available nutrient content of the different biochar-based and peat-based growing media at the beginning of the three growing seasons (barley, wheat, and maize, left columns) and after the plant growth experiment (right columns).

### 3.3. Soil Organic Matter Stability and Quality

The total amount of soil organic carbon ranged between 100 and 200 g, with no significant difference between biochar- and peat-containing growing media (Figure 6a). Soil organic carbon stocks decreased in the following order: Terra Magica > Palaterra > Dehner mit Vorratsdünger > Compo Sana ≈ Dehner die Leichte ≈ Sonnenerde. In all but one treatment, soil organic carbon stock decreased by 6−33% during the experiment (Figure 6a). In Dehner mit Vorratsdünger, soil organic carbon stock increased by 13% (Figure 6a).

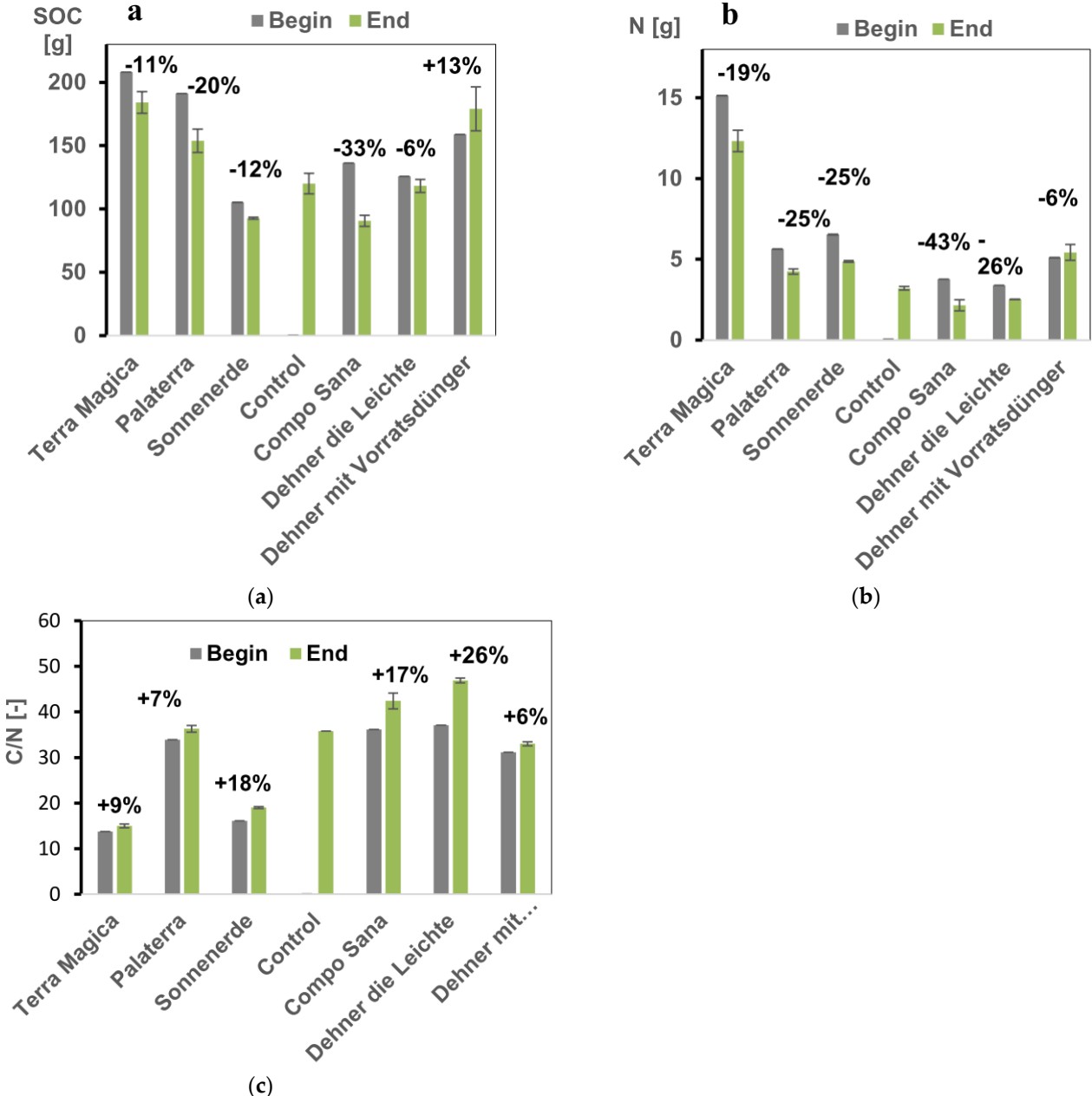

**Figure 6.** Total organic carbon and nitrogen balance of biochar- and peat-based growing media (**a**) Soil organic carbon (**b**) Total nitrogen stock and (**c**) C/N ratio.

The total amount of nitrogen stock ranged between 3 and 15 g, with no significant difference between biochar- and peat-containing growing media (Figure 6b). Nitrogen stock decreased in the following order: Terra Magica > Sonnenerde > Dehner mit Vorratsdünger ≈ Palaterra > Compo Sana ≈ Dehner die Leichte. In all treatments, nitrogen stock decreased by 6–43% during the experiment (Figure 6b).

Carbon to nitrogen ratio ranged between 14 and 37, decreasing in the order Terra Magica ≈ Sonnenerde > Dehner mit Vorratsdünger ≈ Palaterra > Comp Sana > Dehner die Leichte (Figure 6c). In all treatments, carbon to nitrogen ratio increased by 6–26% during the experiment. Apart from Palaterra, biochar-based growing media exhibited a lower carbon to nitrogen ratio compared to peat-based growing media, although during the experiment, the carbon to nitrogen ratio was more homogenous among biochar- and peat-containing media (Figure 6c).

### 3.4. Polycondensed Aromatic Carbon (Polycyclic Aromatic Hydrocarbons and Black Carbon)

Content of the sum of the set of 16 polycyclic aromatic hydrocarbons, proposed by the Environmental Protection Agency for the prevention of health threats, was highest in Terra Magica (0.44 mg kg$^{-1}$), followed by Sonnenerde (0.24 mg kg$^{-1}$), Dehner die Leichte (0.23 mg kg$^{-1}$), Palaterra (0.08 mg kg$^{-1}$), and Compsana (0.07 mg kg$^{-1}$; Table 2). Polycyclic aromatic hydrocarbon content in control and in Dehner mit Vorratsdünger were below detection limit.

**Table 2.** Polycyclic aromatic hydrocarbons (PAH) and black carbon of commercially available peat- and biochar-based growing media used in this study.

| Commercial Growing Media | PAH [1] [mg kg$^{-1}$] | Black Carbon [2] [g kg$^{-1}$] | Black Carbon [3] [g kg$^{-1}$ TOC] | B6CA/BPCA [4] [%] |
|---|---|---|---|---|
| **Peat-Based** | | | | |
| Compo Sana | 0.07 | 20.1 | 46 | 38 |
| Dehner die Leichte | 0.23 | 27.9 | 60 | 35 |
| Dehner mit Vorratsdünger | BDL [5] | 27.2 | 66 | 35 |
| **Biochar-Based** | | | | |
| Palaterra | 0.08 | 46.6 | 126 | 45 |
| Sonnenerde | 0.24 | 7.3 | 123 | 37 |
| Terra Magica | 0.44 | 19.6 | 108 | 43 |

[1] Content of the sum of 16 priority polycyclic aromatic hydrocarbon suggested by the Environmental Protection Agency. For more experimental details, see methods section. [2] Content of polycondensed aromatic carbon measured as sum of benzene polycarboxylic acids, referred to growing media weight. For more experimental details, see methods section. [3] Content of polycondensed aromatic carbon measured as sum of benzene polycarboxylic acids referred to total organic carbon. For more experimental details, see methods section. [4] Relative contribution of highly condensed aromatic moieties within black carbon. For more experimental details, see methods section. [5] Below detection limit.

The black carbon content of biochar-containing growing media ranged between 10 and 50 g kg$^{-1}$ in the order following order: Palaterra > Terra Magica > Sonnenerde (Table 2). Surprisingly, peat-based growing media also contained more or less uniform black carbon content of around 30 g kg$^{-1}$ (Table 2). The contribution of black carbon to total organic carbon (TOC) was more homogeneous across the growing media under study, ranging between 100 and 120 g kg$^{-1}$ TOC, and between 30 and 60 g kg$^{-1}$ TOC for biochar- and peat- based growing media, respectively (Table 2). The pattern of individual benzene polycarboxylic acids exhibited a contribution of 30–45% highly condensed aromatic moieties (Table 2).

### 3.5. Microbial Residues (Amino Sugars)

Total amino sugars content ranged between 0–7 g kg$^{-1}$ growing media, and decreased in the following order: Terra Magica ≈ Dehner mit Vorratsdünger > Palaterra > Sonnenerde > Compo Sana > Dehner die Leichte > control (Figure 7a). In most cases, individual amino sugar content decreased in the following order: glucosamine > galactosamine > muramic acid > mannosamine. Only Compo Sana and Dehner mit Vorratsdünger exhibited higher muramic acid content than individual amino sugars (Figure 7a). The contribution of amino sugars to TOC ranged between 0 –40 g kg$^{-1}$ TOC, decreasing in the following order: Terra Magica > Sonnenerde > Dehner mit Vorratsdünger >> Palaterra > Compo Sana > Dehner die Leichte > control (Figure 7b). Two out of the three biochar-containing growing media (Terra Magica and Sonnenerde) exhibited a higher amino sugar contribution to TOC, as compared to peat-based growing media (Figure 7b).

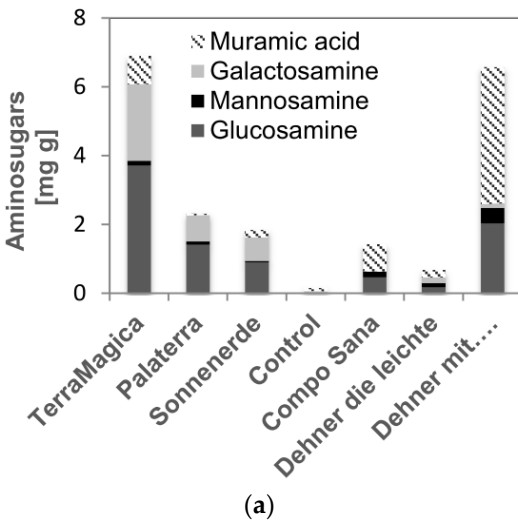
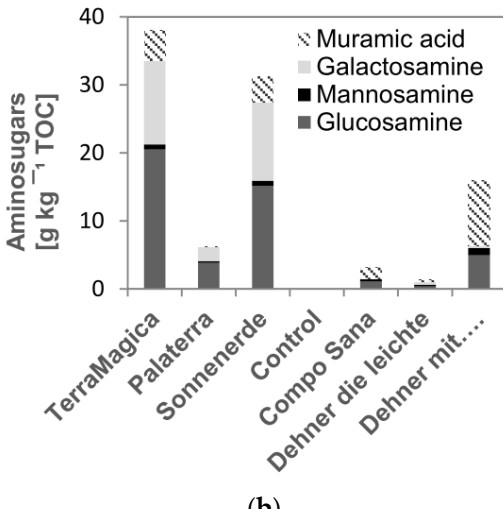

**Figure 7.** Amino sugars content and distribution in biochar- and peat-based growing media (**a**) referred to soil (**b**) referred to soil organic C.

Glucosamine to muramic acid and galactosamine to muramic acid ratios ranged between 0.5–24.3 and 0–13.1, respectively (Table 3). Generally, biochar-based growing media exhibited higher glucosamine to muramic acid and galactosamine to muramic acids ratios, as compared to peat-based growing media (Table 3). Apart from Palaterra, glucosamine to muramic acid ratio and galactosamine to muramic acid ratio were rather homogeneous among biochar- and peat-containing growing media (Table 3).

**Table 3.** Amino sugar ratios of glucosamine to muramic acid and galactosamine to muramic acid in the biochar- and peat based growing media under study.

| Growing Media | GlcN/MurAc [1] | GalN/MurAc [2] |
| --- | --- | --- |
| Compo Sana | 0.58 | 0.01 |
| Dehner die Leichte | 0.79 | 0.72 |
| Dehner mit Vorratsdünger | 0.51 | 0.03 |
| Palaterra | 24.30 | 13.01 |
| Sonnenerde | 3.95 | 3.01 |
| Terra Magica | 4.58 | 2.76 |

[1] Ratio of glucosamine (GlcN) to muramic acid (MurAc).　[2] Ratio of galactosamine (GalN) to muramic acid (MurAc).

## 4. Discussion

### 4.1. Biochar Effect

The effect of biochar is the subject of a huge number of studies, because this charred organic material with polycondensed aromatic and heteroaromatic moieties is the key factor for the stability of soil organic matter in Terra Preta [9]. In our study, the black carbon content of biochar-containing growing media ranged between 10 and 50 g kg$^{-1}$ in the following order: Palaterra > Terra Magica > Sonnenerde (Table 2). The black carbon contribution to total organic carbon ranged between 10 and 20% (Table 2). Mean black carbon content in Terra Preta and natural black soils (chernozems) ranged between 7–17 and 2–10 g kg$^{-1}$, respectively, with a contribution to total organic carbon of about 20% both in Terra Preta [10] and chernozems [11]. The black carbon contents of Sonnenerde and Terra Magica were similar to those of Terra Preta, while Palaterra exhibited a much higher black carbon content of about 50 g kg$^{-1}$ (Table 2).

Surprisingly, peat-based growing media also contained very high black carbon content of about 30 g kg$^{-1}$, but black carbon contribution to total organic carbon was much lower due to the high organic matter content of peat (Table 2). This cannot be explained by

the genesis of peat bogs, which is a gradual layering of about 1 mm per year of whole plants, especially *Sphagnum*, *Eriophorum*, and *Erica* species in an acidic, wet, and anaerobic environment, but it is not a charring process [1]. However, many peats have been burnt prior to harvesting. Peatlands are the soil organic matter-rich ecosystems most affected by fire [12] and, at the beginning, drainage peatlands are dominated by long-lasting smoldering, which can be explained as a flameless, slow, and low temperature combustion of organic matter on the surface and subsurface of the soil [13]. One detection product of peatland fires is black, charred residues from the incomplete combustion of biomass, and the results of charcoal records in sediment and peat cores have been used in recent studies of ecosystems change and fire frequency during the Holocene [14]. Additional studies reported about peatland fires after dewatering in Poland in 2002, Asia in 1997/98, and in Northeast Europe for long periods of time (e.g., months and years) despite rain or weather changes [13]. Other studies reported a non-pyrogenic source of black carbon-like molecules that can be obtained photochemically from dissolved organic matter in peats [15] or by biological production in natural fire-free soils [16].

The contribution of highly condensed aromatic carbon, indicated by mellitic acid (B6CA) to total benzene polycarboxylic acids (BPCA), ranged in biochar- and peat-based growing media between 30 and 45%, with higher values in biochar-containing growing media compared to peat-containing growing media. This indicates that the degree of condensation of biochar is higher than of peat. It was reported that the maximum number of carboxyl groups, especially mellitic acid, decreased the degradation of biochar, but the experiment duration was too short for any alteration of benzene polycarboxylic acids (BPCA) structure or degree of condensation to be measured [17].

The BPCA pattern confirmed that biochar in every biochar-based growing media was produced by pyrolysis, as indicated by a high B6CA/BPCA ratio (Table 2). Suitable influence on soil properties and plant growth could be reported on the basis of biochar content in the growing media. The manufacturer information indicated biochar content in Terra Magica as 20%, Sonnenerde as 25%, while Palaterra released no information.

The content of the sum of the set of 16 polycyclic aromatic hydrocarbons, proposed by the Environmental Protection Agency for the prevention of health threats, in biochar- and peat-based growing media (Table 2) did not exceed the thresholds of 12 mg kg$^{-1}$. In addition, there was no significant correlation between PAH and black carbon content. This may be explained by the fact that PAH are extracted by non-polar organic solvents from soil, while black carbon is digested, and is therefore more quantitative, especially in black carbon-rich soils. Additionally, PAHs are much smaller molecules than black carbon, and, thus, are more prone to global distillation, while the transport of black carbon is limited to smaller regions, if at all.

### 4.2. Bacterial and Fungal Residues/Activities

We used amino sugars and muramic acid content for fungal and bacterial residues [18]. In the biochar-based growing media, individual amino sugar content decreased in the following order: glucosamine > galactosamine > muramic acid > mannosamine. In the peat-based growing media the order was as follows: muramic acid > glucosamine (Figure 7). The same trends were obtained when referring amino sugar content to total organic carbon in order to eliminate variations of soil organic matter content (Figure 7). Similar amino sugar contributions were reported for various soils around the world [18].

Glucosamine cannot be used as indicator for fungal community alone, and muramic acid is not specific for bacteria in soil [18]. Ref. [8] indicated that the ratio between glucosamine and muramic acid might be an indicator for microbial community structure. This ratio ranged between 4–24 in biochar-based growing media, similar to 4–25 for mineral soil [18,19].

Another alternative for characterization of the fungal-to-bacterial ratio is the ratio of galactosamine to muramic acid, because actinobacteria do not contribute significantly to

the mannosamine and galactosamine content [18]. In our study, this ratio was 0.03–13 in biochar- and peat-based growing media (Table 3).

Based on the above-mentioned ratios, microbial residues in biochar-based growing media are dominated by fungal residues, while peat-based growing media are dominated by bacterial residues (Table 3). The highest fungal residue contribution was found in Palaterra, probably due to the highest biochar content, as indicated by black carbon (Table 2), and also due to the addition of fungi, as indicated in the ingredients list (Table 1).

### 4.3. Dilution Effect and Long-Term Stability of Growing Media

The biochar-based growing media Terra Magica and Palaterra did not show a significant dilution effect of substrate with pure sand on biomass yield (Figure 4). Additionally, biomass yields were not significantly higher than when using pure sand as a control (Figure 4). In the case of Palaterra, this might be due to the fermentation process, which is unique to this biochar substrate. On the other hand, in Sonnenerde and all peat-containing growing media, there is a positive correlation between the amount of substrate used and the total biomass yield (Figure 4). In addition, biomass yields were significantly higher when grown in these substrates (even when diluted), as compared to pure sand (Figure 4). Application of biochar (charcoal produced by pyrolysis of biomass) to infertile Oxisols has shown sustainable improvements in soil fertility [4,20,21] but not on pure sand [22]. A meta-analysis demonstrated a mean crop productivity increase of 15% after application of biochar to soil, but no significant difference between any of the application rates [23]. Possible explanations for the different biomass yields of the individual growing media is their composition as summarized in Table 1. In particular the properties of biochar in biochar-based growing media differ due to different feedstocks, biochar production conditions, and processing conditions such as composting, fermentation, and the mixing of source materials and biochar, due to enrichment with microorganisms and activation of the charcoal with nutrients [24]. The combination of aerobic rotting (composting) and two anaerobic fermentation steps are only practiced by Palaterra in order to produce biochar-containing growing media. In contrast, Sonnenerde and Terra Magica are produced by a composting process with different raw materials (Table 1).

Biochar as a bulking agent in composting improves oxygen availability and stimulates microbial growth and respiration rates [25,26]. Recent studies have shown that fermentation induces a negative priming effect in fresh organic matter because it is stable as long as no oxygen is available. When it was kept in an aerobic atmosphere, the intermediates would be mineralized by microorganisms [9].

Soil organic carbon stock decreased by 6–33%, C/N ratio decreased by 6–26%, and nitrogen stock decreased 6–43%, respectively, with no significance between biochar- and peat-based growing media during the experiment (Figure 6). On this basis, organic matter balance/quality and nutrient values especially in biochar-based growing media depend on the synergisms between compost and biochar, because it is necessary to use the relevant feedstock that represent the stabile soil organic matter in the long-term amelioration of soil properties, since the stable pool decomposes only very slowly [27]. The labile soil organic matter pool, on the other hand, provides easily available sources of food for soil organisms and nutrients for plant growth [26,28]. In this context, biochar-based growing media includes, on the one hand, labile fresh household and green waste (a C/N ratio of 17–20) and, on the other hand, biochar as a stable element that influences the balance of the C/N ratio. The C/N ratio of 37 in Palaterra is rather high, but it is nonetheless still lower than peat-based growing media, with a C/N ratio of 100. The combination of compost feedstock and composting process with biochar has an impact on nutrient retention.

Generally, peat-based growing media exhibited a higher C/N ratio compared to biochar-based growing media (Figure 6c). In addition, the C/N ratio increased in all growing media during the experiment (Figure 6c), indicating a higher loss of nitrogen compared to carbon during the experiment, which was generally higher in the peat-based compared to the biochar-based growing media (Figure 6c).

Apart from Palaterra, biochar-based growing media exhibited greater total nutrient stock compared to peat-containing growing media. However, during the experiment, calcium content decreased in biochar-based growing media and potassium content decreased in peat-based growing media. In this context, several authors have claimed that the high specific surface area of biochar is the reason for its higher nutrient retention. Thus, it reduces leaching processes after the addition of ash and organic fertilizer to biochar [29–31].

## 5. Conclusions

In sum, our comparison of biochar- and peat-based growing media on biomass yields and soil properties demonstrates that:

(i)　After three growing cycles, the nutrient leaching is comparable in both biochar- and peat-based growing media. The only difference is that biochar-based growing media loose more calcium, while peat-based growing media loose more potassium.

(ii)　In Sonnenerde and all peat-based growing media, there was a clear amount yield relation when mixed with sand. All other growing media did not show such a trend when diluted with sand but their yield was not significantly higher than pure sand Palaterra had a negative amount yield relation.

The results of our study indicate that biochar-containing growing media are a potential alternative to peat-based growing media in horticulture. The use of biochar-containing growing media has the advantage over peat-based growing media that it is carbon-negative, while peatland destruction releases huge amounts of carbon dioxide into the atmosphere while destroying sensitive ecosystems at the same time. Further studies should investigate physical and hydrological properties, as well as improve our mechanistic understanding of the different biochar-containing growing media.

**Author Contributions:** B.G. conceived the research and prepared the manuscript. A.A.A.A. conducted the analytical work and prepared a first draft of the manuscript. All authors have read and agreed to the published version of the manuscript.

**Funding:** This research received no external funding.

**Institutional Review Board Statement:** Not relevant.

**Informed Consent Statement:** Not relevant.

**Data Availability Statement:** Original data may be available upon person request by the corresponding author.

**Acknowledgments:** We acknowledge analytical support by Heike Mannicke.

**Conflicts of Interest:** The authors declare no conflict of interest.

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
