# Peer review of "Plant Growth and Chemical Properties of Commercial Biochar- versus Peat-Based Growing Media"

_horticulturae, doi:10.3390/horticulturae8040339_

Round 1
Reviewer 1 Report
Review: Comparative performance of commercial biochar- and peat- based growing media for soil fertility and plant growth. HORTICULTURAE
In this paper authors compare several growing media (peat and biochar-based) in terms of plant growth and soil quality (nutrients, microbiological activity, organic matter stability, polycyclic aromatic hydrocarbons, etc.). The subject is interesting and of evident practical interest, but there are some aspects that need improvement. Specifically, introduction needs more background; discussion could be shortened through avoiding listing again the results and should instead focus on explaining the general trends observed. There is also a great number of methodological and editing errors.
Please find here some comments that I hope will be helpful to improve the manuscript.
- Introduction must provide deeper background about:
- The opportunity/reasons of using cereals in this experiment.
- Why the methods/parameters to evaluate the suitability of growing media were chosen, in particular the contents of polycondensed aromatic carbon and its relationship with soil properties or plant growth.
- It is not clear what the authors mean with the objective “ to assess whether there is any amount effect”
- There is some information missing in materials and methods section:
- As shown in figures, control treatment is pure sand (or growing media diluted as 0%). This have to be clearly stated in lines 65-68, because the sentence “…was diluted with sand (0,25,50,75, 100 %)” is confusing.
- The experimental design is unclear: Three separate experiments (one per plant species) or one experiment with three growing seasons (i.e. one plant after another). For the first option I guess 2 substrates x 4 dilutions plus a common control x 5 replicates x 3 plant species = 135 pots. For the second option it would be different. In other words, What do you mean with two/three growing periods (lines 389, 448).
- What about H8 in table 1 footnote?
- What yield expressed as biomass per plant or as biomass per pot?
- Data about height and number of plants per pot (line 79) are not shown in tables/figures.
- What about heavy metals determination (summary, line 17)?
- I find description of method for BC and amino sugars too detailed, maybe they can be summarized.
- Please explain more in detail in M&M about EPA PAK
- Spearman correlation test (line 167) or Pearson correlation (lines 196, 210, 223) ?
- Data presentation could be improved:
- Indicate how plant biomass is expressed (per plant, pot,…) in figures 1-3 and the corresponding text.
- Although it can be read in figure legends, I suggest authors also include biochar/peat in X-axis in figures 1-4. Same comment for figures 5-8, maybe as a letter (B/P for biochar/peat) before the name of the commercial substrate. It will make it easier to interpret for the reader.
- Growth of wheat and maize in Palaterra is similar to that in sand. It is unexpected. It would be interesting to provide a possible explanation
- Please check lines 213-215. Line 217: change demonstrated by produced, led to or something like this.
- Figure 5 is hard to follow, as columns for begin and end are equal in colour. Please modify. Check the figure 5 legend for grammar: change nutrients by soil nutrients; rephrase the last sentence.
- Explain more in detail what 16 EPA PAK are. Do you mean the set of 16 Polycyclic aromatic carbon proposed for Environmental Protection Agency for the assessment of health threats in environmental samples? Readers would appreciate more information in introduction and M&M section. Give the exact data in text (lines 263-266, and also in other sections) is unnecessary, as they can be seen easily in the corresponding figure.
- I suggest authors move figures 7-8 to a table, because there are just one bar per treatment, some date are below detection limit, some substrate numbers cannot be read.
- Discussion is too long and rather descriptive. Some paragraphs are just a listing of results (for example, lines 310-319, 361-367, 387-394, 418-420, etc.). As I said before, it would benefit from shorten, please keep result listing to a minimum and focus on explanations for the trends observed instead.
- It is expected that substrate containing fungi show higher fungal residues. Discussion in subsection 4.2 must take it into account more clearly.
- Sentence 389-391 is awkward. Please rephrase.
- The second conclusion sounds rare. Conclusions must be write in line with hypothesis, which is comparison among biochar and peat substrates rather than the particular effect of sand dilution.
- Abstract needs language revision. Some tips: It is not the Sonnernede substrate which showed increased yield, but the plants growing in it. Lines 20-21 are too vague, as no information about results are provided It has been perceived that…..was observed is redundant. Sand cannot be considered a degraded poor soil.
Author Response
Reviewer #1:
In this paper authors compare several growing media (peat and biochar-based) in terms of plant growth and soil quality (nutrients, microbiological activity, organic matter stability, polycyclic aromatic hydrocarbons, etc.). The subject is interesting and of evident practical interest, but there are some aspects that need improvement. Specifically, introduction needs more background; discussion could be shortened through avoiding listing again the results and should instead focus on explaining the general trends observed. There is also a great number of methodological and editing errors.
Authors: We extended introduction and shortened discussion.
Please find here some comments that I hope will be helpful to improve the manuscript.
-
Introduction must provide deeper background about:
-
The opportunity/reasons of using cereals in this experiment.
Authors: We do not think that it is necessary to justify choice of experimental plants. We could have chosen any but decided to use agronomically relevant crops. However, this information is redundant.
-
Why the methods/parameters to evaluate the suitability of growing media were chosen, in particular the contents of polycondensed aromatic carbon and its relationship with soil properties or plant growth.
Authors: Again, the choice of investigation properties was obvious, why there should be a justification for those?
-
It is not clear what the authors mean with the objective “ to assess whether there is any amount effect”
Authors: Amount effect is the amount of growing media as shown in each single figure...
-
There is some information missing in materials and methods section:
-
As shown in figures, control treatment is pure sand (or growing media diluted as 0%). This have to be clearly stated in lines 65-68, because the sentence “…was diluted with sand (0,25,50,75, 100 %)” is confusing.
Authors: Sentence was corrected.
-
The experimental design is unclear: Three separate experiments (one per plant species) or one experiment with three growing seasons (i.e. one plant after another). For the first option I guess 2 substrates x 4 dilutions plus a common control x 5 replicates x 3 plant species = 135 pots. For the second option it would be different. In other words, What do you mean with two/three growing periods (lines 389, 448).
Authors: It is clear what growing periond means, namely one experiment with three subsequent growing seasons to investigate the longer term effect of the growing media.
-
What about H8 in table 1 footnote?
Authors: Was corrected to H7.
-
What yield expressed as biomass per plant or as biomass per pot?
Authors: Biomass per pot. This was added to each corresponding figure.
-
Data about height and number of plants per pot (line 79) are not shown in tables/figures.
Authors: Sentence was deleted.
-
What about heavy metals determination (summary, line 17)?
Authors: Heavy metals was deleted.
-
I find description of method for BC and amino sugars too detailed, maybe they can be summarized.
Authors: Description of methods for black carbon and amino sugars were shortened.
-
Please explain more in detail in M&M about EPA PAK
Authors: Description of EPA PAK was more detailed.
-
Spearman correlation test (line 167) or Pearson correlation (lines 196, 210, 223) ?Authors
Authors: We used Spearman correlation. This was corrected in the Figure legends.
-
Data presentation could be improved:
-
Indicate how plant biomass is expressed (per plant, pot,…) in figures 1-3 and the corresponding text.
Authors: Plant yield is of course given per pot but we included this information in the corresponding figure legends.
-
Although it can be read in figure legends, I suggest authors also include biochar/peat in X-axis in figures 1-4. Same comment for figures 5-8, maybe as a letter (B/P for biochar/peat) before the name of the commercial substrate. It will make it easier to interpret for the reader.
Authors: The figures are clearly readable. On the x-axis, the amount of individual substrates if given and the type of substrates are given with different symbols being displayed with a corresponding figure legend. In addition, the left figure contains peat-based substrates, while the right figure containes the biochar substrates. Therefore, we leave the figures as they are.
-
Growth of wheat and maize in Palaterra is similar to that in sand. It is unexpected. It would be interesting to provide a possible explanation
Authors: The explanation is that Palaterra is not better than pure sand, which was provided in the discussion.
-
Please check lines 213-215. Line 217: change demonstrated by produced, led to or something like this.
Authors: “Demonstrated” was replanced by “exhibited”.
-
Figure 5 is hard to follow, as columns for begin and end are equal in colour. Please modify. Check the figure 5 legend for grammar: change nutrients by soil nutrients; rephrase the last sentence.
Authors: Left is begin, right is end…Nutrients was changed by plant-available nutrients because what we investigated was no soil but growing media. Nevertheless, the ambiguity was the fraction of nturients.
-
Explain more in detail what 16 EPA PAK are. Do you mean the set of 16 Polycyclic aromatic carbon proposed for Environmental Protection Agency for the assessment of health threats in environmental samples? Readers would appreciate more information in introduction and M&M section. Give the exact data in text (lines 263-266, and also in other sections) is unnecessary, as they can be seen easily in the corresponding figure.
Authors: Yes of course. We made this more clear in the material and methods section.
-
I suggest authors move figures 7-8 to a table, because there are just one bar per treatment, some date are below detection limit, some substrate numbers cannot be read.
Authors: Figures 7and 8 was converted to a table.
-
Discussion is too long and rather descriptive. Some paragraphs are just a listing of results (for example, lines 310-319, 361-367, 387-394, 418-420, etc.). As I said before, it would benefit from shorten, please keep result listing to a minimum and focus on explanations for the trends observed instead.
Authors: we tried to reduce repetition of results to a minimum. However, this cannot completely avoided in separate results and discussion sections.
-
It is expected that substrate containing fungi show higher fungal residues. Discussion in subsection 4.2 must take it into account more clearly.
Authors: This was already mentioned in the text (L324ff)
-
Sentence 389-391 is awkward. Please rephrase.
Authors: Sentence was re-phrased.
-
The second conclusion sounds rare. Conclusions must be write in line with hypothesis, which is comparison among biochar and peat substrates rather than the particular effect of sand dilution.
Authors: This refers to the amount effect which was our second hypothesis (L57)
-
Abstract needs language revision. Some tips: It is not the Sonnernede substrate which showed increased yield, but the plants growing in it. Lines 20-21 are too vague, as no information about results are provided It has been perceived that…..was observed is redundant. Sand cannot be considered a degraded poor soil.
Authors: We revised the abstract. With respect to considering sand as as a surrogate for a poor soil, or at least to an adequate control, we stick to this.
Reviewer 2 Report
Dear colleague,
your manuscript brings some inspiring insights. Kindly consider improving its communication:Title:
- condensate the main revelation into a short claim
Abstract:
- better follow the established schema of Abstract: A/ introduction (urgency and significance of the research hypothesis); B/ principles of the methods used + key results; C/ conclusions (commercial and environmental impacts)
- originality of these "discoveries" is questionable, there are thousands of studies from which similar findings can be deduced (many of them mention the same products), better highlight the novelty of your discovery (reveal the mechanisms behind your results)
Introduction:
- this chapter should provide the reader with all the information (chemical, physical and biological interactions of biochar with substrate/plant) necessary to understand the context and issues addressed in the following chapters
- clearly build the research hypothesis (justify its urgency and significance from industrial point of view)
Materials and Methods:
- the methodology must be presented in such a way that it can be reproduced anytime, by anyone, anywhere (do not create obstacles like referring to specific location etc.)
- each material/reactant and apparatus used needs to be presented in detail (serial number, setup, manufacturer, country of origin, purity etc.)
Results:
- each Tab. and Fig. should be provided with caption that describes A/ what can be seen and B/ how is this relevant to the research hypothesis
Discussion:
- show more criticism to your methods (can the declarations made by manufacturers be fully trusted? What are the weaknesses of the methods used? Where do the main measurement inaccuracies arise? What are the limitations from a commercial point of view? what about availability of nutrients to plant growth?)
- much has been written on the physical, chemical and biogical aspects associated with biochar applications, but little has been published on the economics (provide some comments of competitiveness with existing products)
- propose some improvements and direction for future research
- kindly note that C/N has been repeatedly and independently confirmed as an erroneous analysis in the last century because it does not reflect the availability of nutrients to living organisms
- consider providing deeper synthesis above these results, what are the driving mechanisms?
Conclusions:
- please understand that the Conclusion chapter is not a sum of results, present only original revelations that have the potential to expand the horizon of human knowledge
Author Response
Reviewer #2:
Dear colleague,
your manuscript brings some inspiring insights. Kindly consider improving its communication:
Title:
-
condensate the main revelation into a short claim
Abstract:
-
better follow the established schema of Abstract: A/ introduction (urgency and significance of the research hypothesis); B/ principles of the methods used + key results; C/ conclusions (commercial and environmental impacts)
-
originality of these "discoveries" is questionable, there are thousands of studies from which similar findings can be deduced (many of them mention the same products), better highlight the novelty of your discovery (reveal the mechanisms behind your results)
-
Authors: We thank reviewer #2 for this valuable hints. However, there are no similar studies using the same commercial substrates in such a sophisticated study. In addition, our approach was related to practical issues (plant production performance) rather than elucidating mechanistic effects behind it. Therefore, such endeavor is impossible with our study design
Introduction:
-
this chapter should provide the reader with all the information (chemical, physical and biological interactions of biochar with substrate/plant) necessary to understand the context and issues addressed in the following chapters
-
Authors: We extended the biochar effects state of the art in this section.
-
clearly build the research hypothesis (justify its urgency and significance from industrial point of view)
-
Authors: Was clearly stated (L58)
Materials and Methods:
-
the methodology must be presented in such a way that it can be reproduced anytime, by anyone, anywhere (do not create obstacles like referring to specific location etc.)
-
Authors: This is contrary to reviewer #1, who requested a shortening of our methods section, which we followed.
-
each material/reactant and apparatus used needs to be presented in detail (serial number, setup, manufacturer, country of origin, purity etc.)
-
Authors: see previous comment.
Results:
-
each Tab. and Fig. should be provided with caption that describes A/ what can be seen and B/ how is this relevant to the research hypothesis
Discussion:
-
show more criticism to your methods (can the declarations made by manufacturers be fully trusted? What are the weaknesses of the methods used? Where do the main measurement inaccuracies arise? What are the limitations from a commercial point of view? what about availability of nutrients to plant growth?)
-
Authors: These aspects were discussed now in more detail.
-
much has been written on the physical, chemical and biogical aspects associated with biochar applications, but little has been published on the economics (provide some comments of competitiveness with existing products)
-
Authors: This is beyond the scope of our study (plant performance)
-
propose some improvements and direction for future research
-
kindly note that C/N has been repeatedly and independently confirmed as an erroneous analysis in the last century because it does not reflect the availability of nutrients to living organisms
-
Authors: Nevertheless, the C/N ratio is a valuable measure for soil organic matter quality and is a practial measure for at least the availablity of nitrogen, which is still current paradigm.
-
consider providing deeper synthesis above these results, what are the driving mechanisms?
-
Authors: As our study has no mechanistic design / intention, such discussion would be too much speculation. We included a sentence at the end of the conclusions, that further studies are necessary to investigate the mechanistic understanding of our results.
Conclusions:
- please understand that the Conclusion chapter is not a sum of results, present only original revelations that have the potential to expand the horizon of human knowledge
Reviewer 3 Report
The use and testing of soil improvers already on the market is an interesting and promising topic. It is very important for consumers to have data on the comparative analysis of such products in the market.
The laboratory experiment presented by the authors is quite interesting and original.
In the course of studying the submitted manuscript, some questions arise for researchers:
- Has a comparison been made of the degree of aggregation of commercial products based on peat and biochar. If the product was mixed with sand, then the ability to form soil aggregates in peat and biochar will be different, which means that the ability to create a favorable water-air environment in the root zone for plants will also differ. It would probably be possible to add a measurement of the filtration rate of the resulting experimental mixtures or their ability to hold water.
- In the experiment, you determined the content of different forms of nitrogen. It would be better focus on forms of nitrogen available to the plant, and in the description to compare the increase or consumption of this element during the growing season, and not the initial and final values of the content of these elements.
- The formation of the humus system of the soil occurs over a very long period. Is it possible that in such a short period in your experiments, organic carbon became part of the humus system? If we are talking about the reserves of organic carbon and the rate of mineralization, then it is better to calculate these indicators in a year or two, when a full-fledged humus system is formed in the formed soil mixture.
- Have you information about the content of humic and fulvic acids in peat mixtures? If yes, then it would be nice to add them to the work. Mixtures of peat and biochar may differ from each other in this parameter.
- It makes sense that you would add sand as an all-purpose neutral, but if you recommend the commercial biochar products you have tested in horticulture and agriculture, will the same results be obtained when mixed with soil?
- Laboratory experiment creates stable climatic conditions. Can you guess in what climates and soils you can benefit from the commercial biochar products you are testing?
- It would be nice to add information on the rate of decomposition of peat and biochar as evidence of the benefits of these commercial products.
- The formation of experimental mixtures based on components with an active microbial component and fungal mycorrhiza can also explain some of the results of your experiment.
I hope that these questions and comments will improve your article.
Author Response
Reviewer #3:
The use and testing of soil improvers already on the market is an interesting and promising topic. It is very important for consumers to have data on the comparative analysis of such products in the market.
The laboratory experiment presented by the authors is quite interesting and original.
In the course of studying the submitted manuscript, some questions arise for researchers:
-
Has a comparison been made of the degree of aggregation of commercial products based on peat and biochar. If the product was mixed with sand, then the ability to form soil aggregates in peat and biochar will be different, which means that the ability to create a favorable water-air environment in the root zone for plants will also differ. It would probably be possible to add a measurement of the filtration rate of the resulting experimental mixtures or their ability to hold water.
Authors: During the short time of the experiment, it is most unlikely that the substrates formed any aggregates with the added sand at least not in pedogenic sense. In addition, measurement soil physical properties is beyond the scope of our study. However, we added a sentence at the end of the conclusions that such measurements might be interesting for researchers and growing media industry.
-
In the experiment, you determined the content of different forms of nitrogen. It would be better focus on forms of nitrogen available to the plant, and in the description to compare the increase or consumption of this element during the growing season, and not the initial and final values of the content of these elements.
Authors: We did not investigate different forms of nitrogen. Again, although interesting, this was beyond the scope of our study.
-
The formation of the humus system of the soil occurs over a very long period. Is it possible that in such a short period in your experiments, organic carbon became part of the humus system? If we are talking about the reserves of organic carbon and the rate of mineralization, then it is better to calculate these indicators in a year or two, when a full-fledged humus system is formed in the formed soil mixture.
Authors: Organic carbon is part of “humus”. We did three subsequent plant growth cycles being much more than such substrate normally are used. As you can see, especially for the peat-containing substrates, there was a tremendous loss of total organic carbon, being caused by mineralization of soil organic matter.
-
Have you information about the content of humic and fulvic acids in peat mixtures? If yes, then it would be nice to add them to the work. Mixtures of peat and biochar may differ from each other in this parameter.
Authors: We do not have such information and scientifically, this information is useless as there is only limited scientific value of this operationally defined organic matter fractions.
-
It makes sense that you would add sand as an all-purpose neutral, but if you recommend the commercial biochar products you have tested in horticulture and agriculture, will the same results be obtained when mixed with soil?
Authors: Mixing with soil depends on the properties of the soils used (physical, chemical, biological). Sand was only used as a surrogate for the most infertile “soil” and the purpose of this approach was to see if there is any amount effect of the substrates as mentioned in the text already.
-
Laboratory experiment creates stable climatic conditions. Can you guess in what climates and soils you can benefit from the commercial biochar products you are testing?
Authors: No because we did not test climate as variable.
-
It would be nice to add information on the rate of decomposition of peat and biochar as evidence of the benefits of these commercial products.
Authors: This information is given in Figure 6 (Loss of TOC over the course of the experiment).
-
The formation of experimental mixtures based on components with an active microbial component and fungal mycorrhiza can also explain some of the results of your experiment.
Authors: Yes of course.
I hope that these questions and comments will improve your article.
Authors: Thank you so much for your valuable hints and comments.
Round 2
Reviewer 1 Report
Review 2: Comparative plant growth and chemical properties of commer-2 cial biochar- and peat-based growing media
Authors have addressed most of the suggestions but there are still some minor issues remained that can be easily corrected. I honestly think they will improve this interesting manuscript.
- As the observed effects may considerably vary depending on the test plant used I suggest authors include at least that “several cereals were chosen in this experiment as agronomically relevant crops” and briefly outline the opportunity for it (in introduction or methods and in summary –line 15).
- Summary, line 28: It sounds as considering sand as soil. Although I understand author´s reason to maintain the sentence, I still suggest deleting or rephrasing “degraded soil such as sand” .
- Experimental design is now clearer, but please provide total number of pots.
- Growth of wheat and maize in Palaterra is similar to that in sand. It is unexpected. Although it is stated in discussion, readers would appreciate an in-depth consideration or explanation.
5. My main concern is: Discusssion and conclusions have only been slightly modified. As also reviewer 2 suggest, these sections should avoid listing again the results. I suggest more-in depth revision.
Author Response
Review 2: Comparative plant growth and chemical properties of commercial biochar- and peat-based growing media
Authors have addressed most of the suggestions but there are still some minor issues remained that can be easily corrected. I honestly think they will improve this interesting manuscript.
-
As the observed effects may considerably vary depending on the test plant used I suggest authors include at least that “several cereals were chosen in this experiment as agronomically relevant crops” and briefly outline the opportunity for it (in introduction or methods and in summary –line 15).
Authors: Done.
-
Summary, line 28: It sounds as considering sand as soil. Although I understand author´s reason to maintain the sentence, I still suggest deleting or rephrasing “degraded soil such as sand” .
Authors: We re-phrased the sentence to:
Overall, our results indicated that biochar-containing growing media, especially Sonnenerde is a potential alternative for peat-based growing media in horticulture and can enhance degraded soils.
-
Experimental design is now clearer, but please provide total number of pots.
Authors:
In total, 45 polyvinylchlorid pots (15 cm in diameter and 40 cm in height) were supplied with drainage holes at the bottom (L72).
-
Growth of wheat and maize in Palaterra is similar to that in sand. It is unexpected. Although it is stated in discussion, readers would appreciate an in-depth consideration or explanation.
Authors:
Although this is speculation, we included a sentence that this might be due to the fermentation process which is unique in this biochar substrate (L345).
-
My main concern is: Discusssion and conclusions have only been slightly modified. As also reviewer 2 suggest, these sections should avoid listing again the results. I suggest more-in depth revision.
Authors: We deleted repetition of results as much as possible. Again, an in-depth explanation of results is not possible because this was no mechanistic study but a more or less practical comparison of the agronomic performance of peat- vs. biochar-based substrates. Therfore, in order to avoid to speculative interpretation of obtained results / differences, we prefer to keep this already modified discussion as it is now, unless the reviewer has specific hints about where and how exactly further scientifically sound interpretations are possible.